# TRAF1 from a Structural Perspective

**DOI:** 10.3390/biom14050510

**Published:** 2024-04-23

**Authors:** Hyunseok Jang, Subin Kim, Do Yeon Kim, Ju Hee Han, Hyun Ho Park

**Affiliations:** College of Pharmacy, Chung-Ang University, Seoul 06974, Republic of Korea; john53031112@naver.com (H.J.); subinkim0814@daum.net (S.K.); doyun0509@naver.com (D.Y.K.); gkstlrkans4@naver.com (J.H.H.)

**Keywords:** protein interaction, TRAF1, TRAF domain, structure

## Abstract

Tumor necrosis factor receptor-associated factor (TRAF) proteins play pivotal roles in a multitude of cellular signaling pathways, encompassing immune response, cell fate determination, development, and thrombosis. Their involvement in these processes hinges largely on their ability to interact directly with diverse receptors via the TRAF domain. Given the limited binding interface, understanding how specific TRAF domains engage with various receptors and how structurally similar binding interfaces of TRAF family members adapt their distinct binding partners has been the subject of extensive structural investigations over several decades. This review presents an in-depth exploration of the current insights into the structural and molecular diversity exhibited by the TRAF domain and TRAF-binding motifs across a range of receptors, with a specific focus on TRAF1.

## 1. Introduction

Tumor necrosis factor (TNF) receptor-associated factor (TRAF) proteins serve as vital signaling mediators in diverse cellular pathways, notably in immune responses and apoptosis. They achieve this by binding to a range of cellular receptors, such as tumor necrosis factor receptor (TNF-R), interleukin 1 receptor, Toll-like receptor (TLR), nucleotide-binding oligomerization domain-like receptor (NLR), and cytokine receptors [1,2,3,4]. TRAF protein comprises seven family members, TRAF1 to TRAF7, in mammals. One hallmark feature of TRAF family proteins, with the exception of TRAF1, is the presence of a homologous RING domain located at the N-terminus. This RING domain, reminiscent of those observed in various E3 ubiquitin ligases, serves as the central component of the ubiquitin ligase catalytic domain, playing a pivotal role in facilitating ligase activity [5,6]. Furthermore, except for TRAF7, TRAF family members harbor a TRAF domain at the C-terminus, pivotal for interacting with diverse receptors and comprising approximately 230 amino acid residues (Figure 1A). This domain can be further divided into two distinct subdomains: the N-terminal coiled-coil domain (TRAF-N) and the C-terminal globular domain (TRAF-C). In solution, TRAF family proteins assemble into a trimeric structure reminiscent of a mushroom, driven by the interactions mediated through the TRAF domain, which acts as the functional unit essential for TRAF signaling [7,8]. 

It is generally known that TRAF proteins fulfill dual roles, functioning both as E3 ubiquitin ligases and as scaffolds within cellular signaling pathways. The scaffolding function primarily relies on the TRAF domain, facilitating interactions between various membrane receptors and downstream effector molecules, notably protein kinases such as ASK1, IRAKs, TAK1, and MEKK1 [9,10,11,12]. The E3 ligase activity of TRAFs has undergone extensive investigation, leading to the identification of substrates specific to each family member [6,13,14,15,16]. 

TRAF1 was initially recognized as an adaptor protein interacting with TNFR2, setting it apart as a distinct member of the TRAF family due to its absence of an N-terminal RING domain [17]. Because of this reason, TRAF1 lacks E3 ubiquitin ligase activity. Involvement of TRAF1 in TNFR2-mediated signaling within T cells is established, where it acts as a negative regulator by direct interaction with TRAF2 to inhibit apoptotic cell death of T cells [18,19,20]. In addition to its anti-apoptotic function, a pro-apoptotic role of TRAF1 in a certain cellular context has been also reported in neuronal cells [21]. Furthermore, research has demonstrated that TRAF1 performed a significant role in the pathogenesis of hepatic steatosis and associated metabolic disturbances, suggesting its involvement as a positive modulator of insulin resistance [22]. 

Given their pivotal roles in critical cellular signaling, TRAF proteins are implicated in various human diseases, including metabolic diseases, cancer, and inflammatory diseases, making them potential drug targets [23,24,25,26,27]. Consequently, structural studies on the TRAF family were explored in the early 2000s to clarify the mechanisms by which specific TRAF proteins interact with diverse receptors via constrained binding interfaces and to understand how similar binding interfaces among TRAF family members discern their particular binding partners. The structures of the TRAF domain within TRAF2, TRAF3, and TRAF6, as well as their complexes with various receptors, were initially solved through X-ray crystallography in the late 1990s and early 2000s, marking significant milestones in understanding their molecular interactions [28,29,30]. Conversely, the structural characterization of TRAF1, TRAF4, and TRAF5, along with their receptor complexes, occurred more recently, allowing for a broader and deeper understanding of the diverse roles played by TRAF family members in cellular signaling pathways [31,32,33,34,35,36]. This review intends to explore the current knowledge surrounding the structural and molecular diversity within the TRAF domain, with a special focus on the TRAF1 family, based on available structural information.

**Figure 1 biomolecules-14-00510-f001:**
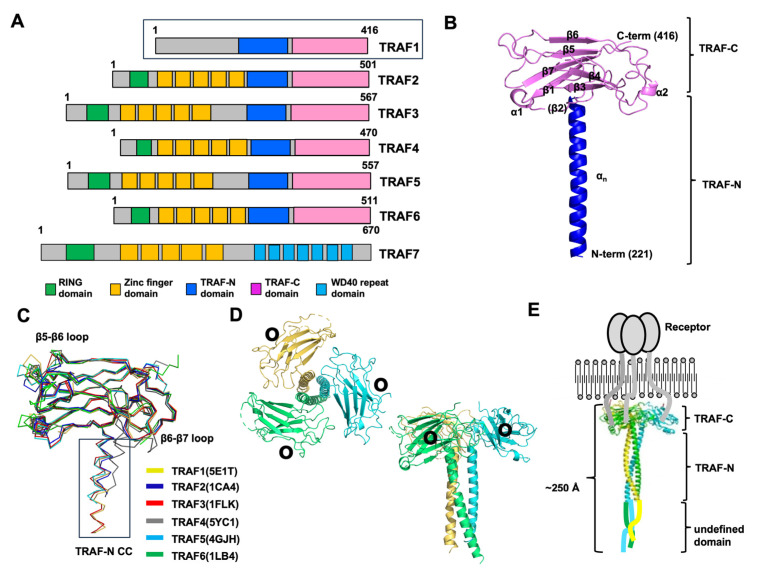
An overview of the various aspects of the TRAF family structure focusing on TRAF1. (**A**) The domain boundary in the TRAF family, showing the regions where specific functional domains are located. (**B**) Representative cartoon figure showing the monomeric TRAF domain. The structure of TRAF1 (PDB ID: 5E1T) was used as a representative example [31]. The color gradient, transitioning from blue to red, illustrates the chain orientation from the N to C terminus, and secondary structures such as helices and sheets are labeled. (**C**) The superposition of the structures of the TRAF domain, likely showing the structural alignment of TRAF domains from different TRAF family members. PDB IDs are presented next to the name of each protein. (**D**) A cartoon of the trimeric TRAF1 TRAF domain. Each chain is shown separately in distinct colors, providing a clear visual representation of the trimeric structure. The illustration offers both top and side views. Within the structure, black circles are utilized to highlight the receptor-binding region, emphasizing its importance in molecular interactions. (**E**) A full-length structural model depicted alongside a schematic representation illustrating the interaction between TRAF1 and its receptor.

## 2. TRAF Domains and Structure

The TRAF domain, which spans approximately 230 amino acids, is a defining feature of TRAF family proteins, with six members (TRAF1–TRAF6) identified in mammals based on this criterion [37]. This domain can be further divided into two subdomains: the TRAF-N domain and the TRAF-C domain. While the TRAF-C domain predominantly interacts with various receptors, the TRAF-N domain is primarily targeted by intracellular signaling molecules. Even though TRAF domains share structural similarities, each TRAF protein demonstrates distinct biological functions owing to its unique repertoire of interacting partners.

The exploration of TRAF domain structures began with the unveiling of the TRAF2 TRAF domain structure by Dr. Wu’s group in 1999 [28], followed by the characterization of the TRAF6 TRAF domain structure by the same group three years later [29]. Subsequently, the structures of the TRAF domains of TRAF3 [35], TRAF5 [35], TRAF4 [32,33,34], and TRAF1 [31] were also reported. The structural investigation revealed that the TRAF-N domain adopts a coiled-coil structure, whereas the TRAF-C domain was a globular fold consisting of seven to eight anti-parallel β-strands (Figure 1B). When aligning the structures of all six TRAF family members, it becomes apparent that the TRAF-C domain remains highly conserved, while the TRAF-N domain’s position and length vary across TRAF proteins (Figure 1C). Sequence comparison further supports this observation, revealing variability in the length of the TRAF-N domain across the family. Specifically, TRAF1, TRAF4, and TRAF6 exhibit relatively shorter TRAF-N domains, whereas TRAF3 and TRAF5 harbor relatively longer TRAF-N domains. While the overall structures of TRAF family members are highly similar, noticeable structural variations exist. For instance, certain loops within the TRAF domain of TRAF4 and TRAF6 exhibit differences in length and position compared to other TRAF proteins (Figure 1C). Specifically, in TRAF4, the loops connecting β5–β6 and β6–β7 within the TRAF domain are relatively elongated compared to those observed in other TRAF members. Additionally, the TRAF-N coiled-coil domain is uniquely situated solely in the outer layer in TRAF4 (Figure 1C). These structural disparities among TRAF family members likely contribute to their functional divergence. When in solution, the TRAF domain assembles into a stable and functionally significant trimer, taking on a characteristic mushroom-like structure. Specifically, the TRAF-C domain forms the cap, while the TRAF-N coiled-coil domain constitutes the stalk (Figure 1D) [7,31]. Through structural and biochemical analyses, it has been uncovered that several key interaction hot spots, notably positioned on β3, β4, β6, and β7 of the TRAF domains, are essential for receptor accommodation (Figure 1D).

Drawing upon the structural insights provided by the RING domain, zinc-finger domain, and trimeric TRAF domain, researchers have devised a model outlining the reconstituted full-length TRAF structure [38]. Since TRAF1 does not contain any distinct domains at the N-terminal part, it is inferred that the N-terminal part of TRAF1 might be a flexible loop-like structure. Therefore, in the case of TRAF1, the TRAF domain, responsible for interactions with trimeric active receptors, assembles into a functional trimer, whereas the N-terminal segment may exhibit flexibility. (Figure 1E). Consequently, the overall length of the entire TRAF1 molecule is estimated to be approximately 250 Å, and its shape resembles a long stool or mushroom.

## 3. Understanding Receptor Interaction with TRAF Proteins

TRAF family members engage in interactions with a variety of receptors and intracellular molecules, encompassing TNFR2, CD30, TRADD, GPVI, and TANK, facilitating important cellular signaling events. Early structural and biochemical investigations focusing on TRAFs, notably TRAF2 and TRAF3, along with their associated receptors, have highlighted the utilization of three key regions, termed binding hot spots, for these interactions (Figure 2A,B) [28,29,39,40]:

Hot spot 1, commonly referred to as the hydrophobic pocket, includes amino acid residues on β4, β5, β6, and β7.

Hot spot 2, identified as the serine finger, is marked by three serine residues (with one serine residue substituted by alanine in TRAF1 located within β6 and the loop connecting β6 and β7.

Hot spot 3, labeled the polar pocket, comprises polar amino acid residues on β3 and the loop connecting β3 and β4.

The high conservation of residues within these three hot spots across TRAF1, TRAF2, TRAF3, and TRAF5 implies a shared receptor specificity, indicating common interaction modes among these TRAF proteins (Figure 2A,B). However, in TRAF4 and TRAF6, the residues within the three binding hot spots show a lack of conservation, although there are some conserved residues in hot spot 1. This divergence suggests that TRAF4 and TRAF6 possess unique attributes, leading to differences in their binding mode and specificity to receptors compared to typical TRAF family members, such as TRAF2 and TRAF3.

## 4. TRAF1 and Its Receptor Recognition

TRAF1 serves as an adaptor molecule involved in regulating the activation of NF-kappaB and JNK pathways [43,44]. Initially recognized as a binding protein for TNF receptor type 2 (TNFR2) [18], the cellular functions of TRAF1 remain less elucidated in comparison to other TRAF family members. While early studies suggested a negative regulatory role for TRAF1 in TNFR2 signaling in T cells through its interaction with TRAF2 [18], recent research has revealed positive regulatory functions for TRAF1 downstream of TNFR2-, CD30-, and LMP1-mediated signaling pathways, often in association with TRAF2 [20,45,46,47]. Despite its emerging roles in various human diseases, including hepatic and cerebral ischemia/reperfusion injury and anaplastic large cell lymphoma [21,48,49], the structure of TRAF1 is the least studied among the TRAF family, and much is still unrevealed. 

The TRAF1 TRAF domain shares 53% sequence identity with TRAF2, 48% with TRAF3, 35% with TRAF4, 45% with TRAF5, and 33% with TRAF6 (Figure 2A). This sequence analysis suggests that the TRAF1 TRAF domain is more closely related to those of TRAF2, TRAF3, and TRAF5 than to those of TRAF4 and TRAF6. Additionally, phylogenetic tree analysis indicates that the TRAF domain of TRAF1 is evolutionarily more conserved than those of TRAF2, TRAF3, and TRAF5 (Figure 2C).

The conventional binding hot spots and interaction mechanisms of TRAF family members, notably TRAF2 and TRAF3, have been extensively investigated. The minimal consensus motif recognized by TRAF2 and TRAF3 in TRAF-binding proteins, including members of the TNF-R family, CD40, and LMP1, was characterized as Px(Q or E)E# [where x represents any amino acid, and # denotes that acidic or polar amino acids are preferred] (Figure 2D). In structural studies of TRAF2 in complex with various peptides, the most conserved amino acid within the TRAF-binding motif was denoted as P0, representing the zero position of the TRAF-binding motif. Following this naming convention, residues within the Px(Q or E)E motif are categorized as follows: P (P-2), x (P-1), Q or E (P0), E (P1), and # (P2). Specifically regarding CD40, residues within its TRAF2-binding motif are delineated as P (P-2), V (P-1), Q (P0), E (P1), T (P2), and L (P3) (Figure 2D). To incorporate the Px(Q or E)E motif, specific residues within hot spot 1 of TRAF2 (namely, F410, L432, F447, F456, and C469) engage in extensive van der Waals interactions with the P residue positioned at the P-2 site. At the primary structural determinant located at the P0 position, either Q or E interacts with residues within hot spot 2 of TRAF2, composed off the serine triad (S453, S454, and S455). Specifically, at position P0, the Q forms hydrogen bonds with all three serine residues, whereas E at position P0 is capable of forming only one hydrogen bond. Additionally, the Glu residue at position P1 participates in an ionic interaction with R393 and establishes a hydrogen bond with Y395 in TRAF2 (Figure 2D). Key residues implicated in the interaction with the Px(Q or E)E motif exhibit high conservation among TRAF1, TRAF2, TRAF3, and TRAF5, with the exception of one serine residue within the serine triad, substituted by alanine in TRAF1 (A369). This conservation implies that TRAF1, TRAF2, TRAF3, and TRAF5 employ a shared mode of interaction involving the Px(Q or E)E motif.

In addition to the predominantly detected major TRAF-binding motif [Px(Q or E)E motif], several structural studies, such as those on the TRAF2–LMP1 [39] and TRAF3–TANK [49] complexes, have identified a minor motif [Px(Q/E)xxD motif]. In the Px(Q/E)xxD motif, the effectiveness of the TRAF interaction relies on the side chains of residues at positions P-2, P0, and P3, whereas in the major binding motif, the interaction involves the side chains of residues at positions P-2, P0, and P1 (Figure 2D,E). 

In the case of TRAF1, the most recently solved TRAF1-TANK complex structure (this is the only complex structure within the TRAF1 family) revealed a new TRAF-binding motif, PxQxT motif [42]. The structural analysis of the TRAF1-TANK peptide interaction provides valuable insights into the molecular mechanisms underlying their binding. Despite expectations of the involvement of residues at positions P-2, P0, and P3 in the interaction with TRAF1 based on the minor TRAF-binding consensus motif, the complex structure revealed the participation of residues at positions P0, P1, and P2. Specifically, the side chain of Q at the P0 position formed a hydrogen bond with S368 of TRAF1, while residues C at P1 and T at P2 formed hydrogen bonds with D314 of TRAF1 (Figure 2F) [42]. Surprisingly, the residue at P3 (D), which was expected to participate in the interaction with R308 and Y310 of TRAF1, did not directly engage in the interaction in the observed structure. This observation implies that the side chain of D at the P3 position could not participate in the interaction as initially anticipated. Additionally, structural comparison revealed flexibility in the serine triad-containing loop of TRAF1, which moved closer to the TANK peptide upon binding. This movement was attributed to the formation of a hydrogen bond between P0:Q on the TANK peptide and S368 on TRAF1, highlighting the dynamic nature of the TRAF1-TANK interaction and providing insights into the structural changes accompanying peptide binding. This case of interaction was also detected in TRAF3 interaction studies [30,50,51]. Within the PxQxT motif, residue T (P2) binds to a conserved aspartic acid residue (D314 in TRAF1 and D464 in TRAF3). Meanwhile, the interaction modes of P (P-2) and Q (P0) mirror those observed in the major binding motif. Notably, in the minor consensus motif PxQxxD, the presence of D does not significantly impact the interaction with TRAF3.

In the case of TRAF4 and TRAF6, distinct binding motifs have been identified. The TRAF4-binding motif was elucidated in a study involving a TRAF4 complex with two platelet receptors, GPIb and GPVI [36]. According to this structural analysis, the TRAF4-binding motif spans from position P-3 to P0 and is characterized as R-L-X-A, where X can represent any amino acid and A can be substituted with a small uncharged residue. Conversely, the mode of interaction between TRAF6 and receptors has been unveiled through three available complex structures, namely TRAF6–CD40, TRAF6–RANK, and TRAF6–MAVS [51]. The TRAF6-binding motif consists of six amino acid residues, denoted as PxExxZ (where x represents any amino acid and Z signifies an acidic or aromatic amino acid). Small hydrophobic residues can replace P in this motif. Following the conventional labeling system, the nomenclature of this motif is as follows: P (P-2), x (P-1), E (P0), x (P + 1), x (P + 2), and Z (P + 3). Given that the TRAF1 TRAF domain shares only 35% sequence identity with TRAF4 and 33% with TRAF6, it is understandable that TRAF4 and TRAF6 employ different binding strategies compared to those used by TRAF1.

## 5. Concluding Remarks

Our summary provides a comprehensive overview of the present understanding regarding TRAF-binding motifs and their interactions with diverse receptors. It is evident that while TRAF family members share structural similarities, they exhibit distinct binding specificities and preferences for interacting partners. In this discussion, we explored the recently elucidated TRAF1–TANK structure, where the sequence PxQxT emerged as the TRAF1-binding motif. Notably, we observed that Q at position P0 generates a hydrogen bond with the S368 residue of TRAF1, while both C and T at P1 and P2, respectively, engage in hydrogen bonding with D314 of TRAF1.

Recognizing conserved binding motifs like Px(Q/E)E, Px(Q/E)xxD, and PxQxT underscores their significance in facilitating distinct protein–protein interactions. The shared binding hot spots among TRAF1, -2, -3, and -5 enable them to interact with common receptors, highlighting their overlapping roles in certain signaling pathways. Overall, these findings emphasize the dynamic nature of TRAF-mediated signaling and the intricate interplay between TRAF family members and their interacting partners in various cellular processes. Our review will contribute significantly to our understanding of cellular signaling mechanisms and pave the way for the development of targeted therapeutics aimed at modulating TRAF-mediated pathways in various diseases.

As of now, the full-length structure of the TRAF family, including TRAF1, has not been elucidated. The considerable length of TRAF proteins, coupled with their composition of numerous domains interconnected by flexible unstructured loops, likely presents challenges in solving the full-length TRAF structure. However, with the development of advanced structure determination techniques, there is optimism that the full-length TRAF structure, which could significantly enhance our understanding of TRAF-mediated signaling pathways and various receptor recognition processes, will be elucidated in the near future.

## Figures and Tables

**Figure 2 biomolecules-14-00510-f002:**
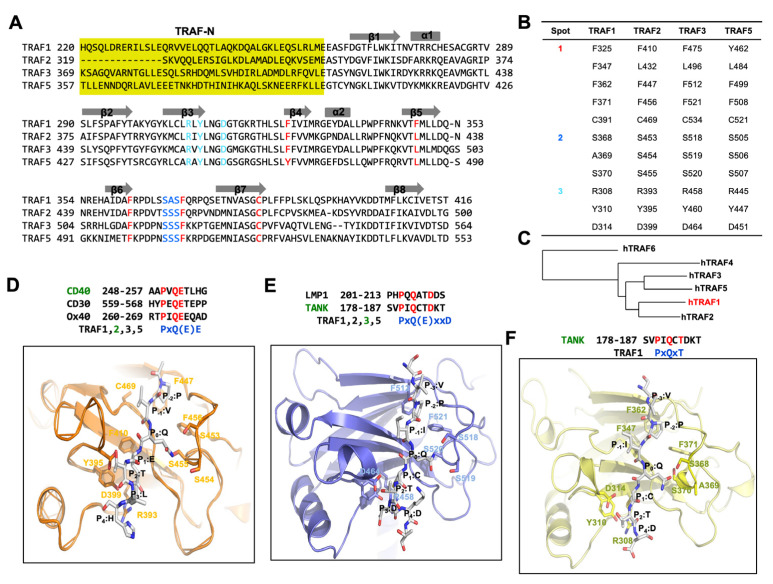
Deciphering the TRAF-binding motif through structural insights from TRAF/receptor complexes, focusing on the case of TRAF1. (**A**) Sequence alignment of TRAF1, TRAF2, TRAF3, and TRAF4 shows the conserved receptor-binding hot spots. The residues involved in the formation of binding hot spots 1, 2, and 3 are colored in red, blue, and cyan, respectively. (**B**) Table summarizing the binding hot-spot-forming residues. (**C**) Phylogenetic tree of TRAF domains from different TRAF families. h indicates human protein. Phylogenetic tree was generated using NCBI tree viewer: https://www.ncbi.nlm.nih.gov/tools/treeviewer (accessed on 25 March 2024). (**D**) A magnified view of a CD40-TRAF2 complex (PDB ID: 1D00) [39]. (**E**) A magnified view of a TANK-TRAF3 complex (PDB ID: 1L0A) [41]. (**F**) A magnified view of a TANK-TRAF1 complex (PDB ID: 5H10) [42].

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
