# Peer review of "TRAF1 from a Structural Perspective"

_biomolecules, 2024, doi:10.3390/biom14050510_

Round 1

Reviewer 1 Report

Comments and Suggestions for Authors

In the review, Jang et. al. summarized structural features of the TRAF-C domains from six human TRAF proteins, TRAF1-6. The authors further compared the different modes adopted by TRAF proteins to recognize receptor sequences, emphasizing the recent discovery that TRAF1 recognizes a slightly different motif in receptors. TRAF1 is unique among the TRAF family for lack of the N-terminal RING domain and zinc fingers. The C-terminal region of TRAF6 consists of the TRAF domain, which can be further divided into TRAF-N and TRAF-C domain. TRAF1 is the founding member of the family. yet its structure was only determined recently in the form of the TRAF1-TANK peptide complex. The authors thoroughly reviewed the structural features of the TRAF-C domains from TRAF1-6, then described the three modes of receptor peptide recognition: a major consensus Px(Q/E)E motif recognized by TRAF1/2/3/5, a minor motif Px(Q/E)xxD, and the most recent PxQxT found in TRAF1 and TRAF3.

The review is well written and clear.

My concerns and comments are below:

1. All citations are unformatted, in the form of “Arkee, 2020 #4854”. Though the citations can be deduced in most cases, it is still difficult to judge whether the authors cited the proper research.

2. I understand that TRAF4 and TRAF6 recognize peptides from proteins other than the TNF-R superfamily. However, it is strange that their structures are not included for comparison.

3. Labeling conserved and similar residues in Figure 2a would help the readers to appreciate the similarities and differences between TRAF1/2/3/5. Alternatively, a heat map of conservation or a phylogenetic tree including TRAF1-6 helps to illustrate the relationship between different TRAF2.

Minor points:

1. In two cases, TRAF1 was misspelled as “TARF1”. 

2. When describing TRAF1:TRAF2 interaction, it is worth citing the heterotrimeric structure of TRAF1:TRAF2 coiled-coil (TRAF-N) domain (Zheng et al. Mol Cell, 2010 PMID: 20385093)

3. Inconsistent usage of “TRAF domain”. Pg. 1, line 31-35, “TRAF domain” is defined as a domain “comprising approximately 230 amino acids”. Pg. 2, line 80, “The TRAF domain, which spans approximately 180 amino acids, …” Clearly the TRAF domain on page 2 is the “TRAF-C domain” as defined on page 1.

Reviewer 2 Report

Comments and Suggestions for Authors

The review written by Jane et al., talks about the current information available in the literature about the molecular and structural diversity in the the TRAF domain and its binding motifs, especially TRAF1. 

Major comments:

  1. The major issue of the manuscript is the absence of citations in the reference section. It was hard to dig into references manually as they were absent in the manuscript section. 
  2. Although the review title suggests that focuses on TRAF1 from a structural perspective, no PDB ID’s were ever mentioned in the text or figures. I would suggest including them as and when the authors have mentioned the use of structures or superpositions in the text. 
  3. The sentence TRAF1 lacks ring domain is duplicated multiple times in the introduction section:
  4. Correct the labelling of the domains along with colors in Fig1. None of the coloring scheme seems to match with the domain architecture. Try to keep the coloring scheme identical to Fig 1 (A-E).
  5. LINE 101: “TRAF4 and TRAF6 exhibit relatively shorter TRAF-N domains, include TRAF1”. TRAF1 seems to be the shortest based on the Fig 1 A.Please clarify or correct the sentence.
  6. Correct spelling mistakes of TRAF1: Line177, 219
  7. Correct the line 195-196 “bonds with all three serine residues, whereas E at position P0 is capable of forming only one hydrogen bond. With serine triad.”
  8. The authors mention that the structure of full-length TRAF1 still not available. I would suggest discussing the reasons which could be responsible for the absence of a full-length structure. Also if you can include Alfafold generated predicted models and the implications of N-terminal 200 residues missing in all crystal structures? 

Round 2

Reviewer 2 Report

Comments and Suggestions for Authors

The authors have tried to address all my comments in the current versionof manuscript.